# Impacts and Predictions of Urban Expansion on Habitat Connectivity Networks: A Multi-Scenario Simulation Approach

Shihui Chang [1], Kai Su [1,2,*], Xuebing Jiang [3], Yongfa You [4,5], Chuang Li [1] and Luying Wang [1]

[1] Guangxi Colleges and Universities Key Laboratory for Cultivation and Utilization of Subtropical Forest Plantation, College of Forestry, Guangxi University, Nanning 530004, China; 2209392003@st.gxu.edu.cn (S.C.); 2209392023@st.gxu.edu.cn (C.L.); 2109302014@st.gxu.edu.cn (L.W.)

[2] State Key Laboratory of Urban and Regional Ecology, Research Center for Eco-Environmental Sciences, Chinese Academy of Sciences, Beijing 100085, China

[3] College of Mechanical Engineering, Guangxi University, Nanning 530004, China; sherryjiang@gxu.edu.cn

[4] International Center for Climate and Global Change Research, College of Forestry, Wildlife and Environment, Auburn University, Auburn, AL 36849, USA; yzy0092@auburn.edu

[5] Department of Earth and Environmental Sciences, Schiller Institute for Integrated Science and Society, Boston College, Chestnut Hill, MA 02467, USA

* Correspondence: sukai_lxy@gxu.edu.cn

**Abstract:** Urban expansion is leading to the loss and fragmentation of habitats, which poses a threat to wildlife. People are hopeful that, through scientific urban planning and the adoption of innovative models for human communities, such a situation can be improved. Thus, a case study was carried out in Nanning City, China, to extract habitats, build an ecological resistance surface, and construct a habitat connectivity network (HCN). To simulate changes to unused land in the future, we put forth the A (the parcel is divided into strips), B (the parcel is divided into two strips), C (the central area of the parcel is planned as a quadrangle), and D (opposite to Scenario C, the peripheral area is green space) scenarios of human communities that guarantee a 30% ratio of green space, and established the corresponding HCNs. The results indicate that: (1) Currently, the habitats cover approximately 153.24 km$^2$ (34.08%) of the entire study area. The ecological corridors in this region amount to a total of 5337, and the topological indicators and robustness indicate a strong stability of the current HCN. (2) With urban expansion, once continuous habitats are being fragmented into smaller green spaces, it is estimated that the habitats will shrink by 64.60 km$^2$. The topological indicators and robustness reveal that the stability of the HCNs becomes lower as well. Multiple scenario simulations demonstrated that Scenario D is better than Scenarios B and C, while Scenario A performed the worst. (3) Furthermore, we observed a stronger negative impact of urban expansion on local connectivity. This indicates that the influence of urban expansion on the local HCNs is often more pronounced and may even be destructive. Our findings can advise urban planners on decisions to minimize the impact of urban expansion on wildlife.

**Keywords:** urban expansion; multi-scenario simulation; habitat connectivity network; topological indicators; robustness

## 1. Introduction

The spread of urban development on unused land in the vicinity of a city can lead to urban expansion. As the global urban population continues to grow, urban expansion continues [1]. In particular, in developing countries like China, urban expansion is still in an accelerated stage of development. Many habitats that were once suitable for wildlife survival have been converted into urban development areas, farmland, or industrial zones, leading to habitat loss and fragmentation [2–4]. Moreover, the disturbance caused by traffic accidents and human activities hinders the movement and migration of wildlife, limiting their ability to find food, reproduce, and avoid predators, which has profound implications for wildlife adaptive processes and subsistence [5–7].

The biodiversity of wildlife in cities is generally low owing to urbanization and human activity, and these wildlife are quite sensitive to environmental changes [4,8–11]. It was discovered that the habitat environment can be efficiently improved by integrating fragmented habitats and constructing a stable habitat connectivity network [12,13]. Ecological networks aim to tackle the threats of habitat fragmentation on species' survival and migration, which have been the focus of ecological research since the 1980s [12–14]. Since Forman and Godron [14] proposed that landscape structure consists of three basic elements (patches, corridors, and matrix), the concept of ecological networks has been extended and developed. Its related exploration and research contribute to the maintenance of biodiversity and promote species dispersal, which has been widely used in academia. In light of this, it is imperative to examine how urban expansion affects habitat connectivity networks in conjunction with the ecological network.

A habitat connectivity network (HCN) is a cohesive entity formed by organically connecting habitat patches through ecological corridors. Habitat patches are specific areas where organisms breed and inhabit, while ecological corridors are vital pathways used to connect these habitat patches [14,15]. The presence of ecological corridors not only maintains connectivity between habitat patches but also provides crucial pathways for species migration and ecosystem services [10,11,15]. However, with ongoing urban expansion, habitat loss and fragmentation have significantly reduced the local habitat area and altered the spatial arrangement of the remaining habitat patches. To address this issue, numerous studies on urban expansion and habitat fragmentation have focused on conducting ecological network assessments, often by constructing ecological networks and using indicators to obtain the required research parameters [16,17]. Among them, one of the most popular techniques for locating ecological corridors is the least-cost path model, which accurately describes the direction of ecological corridors. Connectivity between habitat patches is crucial for maintaining gene flow, ecosystem functionality, and species distribution [18]. By computing topological indices based on HCNs, we can analyze HCNs in depth and develop the corresponding conservation plans. Additionally, these indices help determine the resistance and stability of networks connected by individual nodes. By gaining a deeper understanding of habitat connectivity, we will be able to better maintain the health of ecosystems, facilitate species adaptation and migration, and provide a stronger foundation for sustainable development.

Previous studies have explored the impact of land expansion on connectivity networks, which has been important for understanding the effects of human community development on HCNs for wildlife [11,16]. However, predictions for the future are lacking. Often times, the effects of different scenarios of human community development on HCNs for wildlife are underestimated. Although a considerable amount of research has focused on creating ecological networks, recent research has indicated the importance of analyzing the effects of a variety of land use changes on habitat [16,17,19]. These studies remind us that solely considering the creation of ecological networks may not be comprehensive enough to evaluate the impact of urban expansion on wildlife habitats. However, we have found that case studies on urban expansion in the future are insufficient thus far. Therefore, it is necessary to discuss how urban development influences and constructs HCNs under different circumstances. Such research can help us better understand the impact of urban expansion on wildlife habitats and provide a basis for developing rational urban planning and land use policies. We need more research to model the impact of land use changes on wildlife and determine which scenario of human community development can minimize habitat loss caused by urban expansion. These studies can help decision-makers better understand the needs of wildlife under different development scenarios, promote sustainable development, and protect biodiversity. Therefore, future research should focus on filling the gaps in predicting the impact of urban expansion, strengthening analyses of the impact of human community development on HCNs under different scenarios, and conducting more simulation studies to provide more accurate scientific support for urban planning and land use decision-making. Only then can we minimize the

negative impact of urban development on wildlife habitats and achieve harmony between humans and nature.

This study aims to fill this gap by exploring the impacts of urban expansion on the habitat connectivity network in Nanning City, Guangxi Autonomous Region of southwest China. We also employed a multi-scenario simulation approach to discuss how urban expansion affects and builds HCNs under different scenarios. This is manifested by (1) building a resistance surface and HCNs based on extracting habitat patches; (2) simulating the impact of future urban expansion on the networks; and (3) evaluating the results of the HCNs under different urban expansion scenarios. Thus, we can foresee the impact of urban reshaping processes on wildlife, preserve regional ecological safety, and give theoretical support for the creation of planning schemes by assessing the effects of various scenarios on habitat connectivity.

## 2. Materials and Methods

### 2.1. Study Area

The study area (22°40′15.45″ N–22°54′9.12″ N, 108°9′20.67″ N–108°25′55.21″ E) is in Nanning City District, Guangxi Zhuang Autonomous Region, southern China (Figure 1), which is located south of the Tropic of Cancer and has a subtropical monsoon climate with good water and thermal conditions. The study area is primarily made up of plains and hills, with an altitude range of 4 m to 290 m. Among them, plains are mainly distributed in the northwestern area, and the northeastern and southern areas are mainly hills with evergreen broad-leaved forests as the vegetation cover type. The east–west Yongjiang River runs through the middle of the study area. The soil is fertile and is a place for wildlife to roost, feed, and breed. Most regions include woodland, construction land, and unused land. Unused land can be defined as land that has value for use but is not currently being utilized by people. To accommodate urbanization and population increases while using the least amount of agricultural land as possible, one of the realistic options is to develop unused land. Moreover, the unused land covers 92.28 km$^2$, accounting for 20.5% of the study area. Several of these undeveloped sites in the urban region, will be designed as human neighborhoods. The region is densely populated and much of the unused land will be transformed into human communities. The rapid urban expansion will result in a greater dispersion of habitats, and the spatial pattern of the study area is certain to change significantly [4,7].

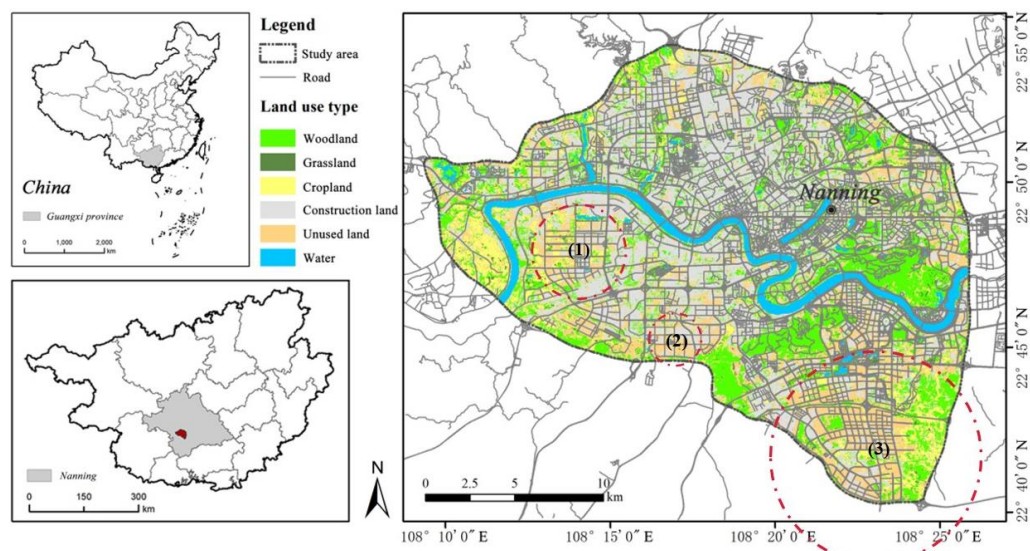

**Figure 1.** Location of the study area. Local study area includes (1) western area, (2) central area, (3) eastern area.

## 2.2. Data Sources

The primary data include digital elevation model (DEM), night lighting, land use, Normalized Difference Vegetation Index (NDVI), population, density of road network, distance from road, slope, density of water network, distance from water, and other related data. Among them, the advanced satellite-based thermal emission and reflection radiometer global digital elevation model (ASTER GDEM) is the source of the DEM. The data for land use were obtained using the Sentinel-2 classification with a resolution of 10 m. In this dataset, land use is divided into six types (cropland, woodland, grassland, water, construction land, unused land). NDVI data were also obtained from Sentinel-2. In this study, the above datasets were converted to the WGS1984 geographic coordinate system with a spatial resolution of 30 m. A detailed description of the datasets and data sources is presented in Table 1.

**Table 1.** Primary datasets presented in this study.

| Dataset | Time | Resolution | Data Sources (Accessed on 1 October 2022) |
|---|---|---|---|
| DEM | 2020 | 30 m | Download from NESSDC (http://www.geodata.cn/) |
| Night lighting | 2021 | 30 m | Download from NESSDC (http://www.geodata.cn/) |
| Sentinel-2 | 2020 | 10 m | Download from USGS (https://earthexplorer.usgs.gov/) |
| Population | 2020 | 30 m | Download from NESSDC (http://www.geodata.cn/) |
| Road | 2020 | 30 m | Download from NESSDC (http://www.geodata.cn/) |
| Water | 2020 | 30 m | Download from NESSDC (http://www.geodata.cn/) |
| DEM | 2020 | 30 m | Download from NESSDC (http://www.geodata.cn/) |

## 2.3. Methods

This study constructed HBNs by selecting habitats and building resistance surfaces. Then, we explored its evolution with future urban expansion, and quantitatively analyze the changes in the spatial distribution of habitats after urban expansion. Notably, changes to urban unused land is the type of land change that will be more variable during future urban expansion. Thus, we considered the study area as a whole, from which we chose three areas with a high density of unused land for local study. In other words, we conducted a study in both overall and local dimensions. The flowchart presents an overview of the methodological steps (Figure 2).

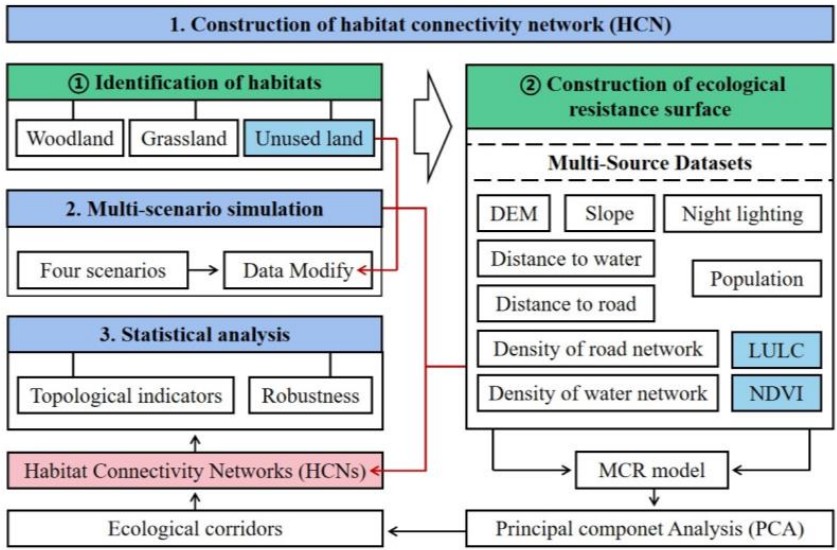

**Figure 2.** Flowchart of the methodological steps.

The first step was to construct the HCN. Woodland, grassland, and unused land were regarded as habitats. Our study combined 10 ecological resistance factors such as DEM, lighting, land cover, NDVI, population, density of road network, distance from road, slope, density of water network, and distance from water [20,21]. The weights of each resistance factor were obtained using principal component analysis, based on which, the minimum ecological cumulative resistance surface was constructed. Combined with the habitats and resistance surface obtained above, we constructed the HCN.

The second step was the multi-scenario simulation. We selected the unused land planned for human communities in urban areas (excluding patches less than 0.006 km$^2$ in area and with a fragmented distribution), and had them change according to four different urban reshaping scenarios. Then, the habitats were selected again and combined with the above-mentioned resistance factors, in which the land cover and NDVI data changed due to the urban expansion. The subsequent operation steps were roughly the same as in the first part, such as calculating the weights of each resistance factor and constructing the minimum ecological cumulative resistance surface. Finally, we compared and analyzed the differences between each resistance factor after urban reshaping.

The third step was the statistical analysis. By calculating and comparing the topological indicators and robustness of the HCNs under different scenarios, the impact of the unused land change on the HCNs during the urban reshaping was determined. The optimum reshaping scenario was finally obtained, and it can be applied to other areas to provide guidance for future planning [17,20,22–25].

### 2.3.1. Selection of Habitats

It is known to all that the protection of wildlife depends on safeguarding and preserving the biological environment of their habitats. Habitats provide sustainable ecosystem services and maintain stable landscape patterns, which are also necessary for wildlife to survive and thrive. The habitats in this research were derived from three main types of land cover, which includes woodland, grassland, and unused land (only for development of human communities). Since small habitats are too dispersed, have weak radiation capacity, and have less impact on the HBNs, we selected land larger than 0.006 km$^2$ as habitats.

### 2.3.2. Circuit Theory and MCR Model

Changes in land cover, such as those resulting from rapid urban expansion, can inhibit or enhance ecological processes to varying degrees. Using complex network theory, ecological processes such as species migration and gene flow are considered electrical currents and landscape resistance are considered resistance, the results of which, can reflect the energy required for species migration or mortality [20,21]. Based on the characteristics of circuit theory, it can offer various methods for calculating the paths taken by ecological processes in the landscape, such as isolating active areas with high levels of ecological activity and creating spatial scenarios of the HCNs in the study area.

When transitioning between habitats, wildlife must overcome a number of obstacles. Several habitats can be connected through ecological corridors, which can provide channels for the interchange of resources and energy. As shown in Table 2, the 10 resistance factors chosen for this study were separated into 5 levels and given the values 1, 2, 3, 4, or 5 in accordance with the resistance's intensity from high to low [20,21]. Therefore, the minimum cumulative resistance surface was built. The specific formula is as follows:

$$R_{mc} = f_{min} \sum_{j=n}^{i=m} D_{ij} \times R_i \qquad (1)$$

where $R_{mc}$ is the value of the minimum cumulative resistance, $f$ is an unidentified negative function reflecting the inverse relationship between the minimum cumulative resistance and the propagation or diffusion process, $R_i$ is the resistance coefficient of the landscape

unit to the propagation and diffusion process, and $D_{ij}$ is the spatial distance between the source $i$ and landscape unit $j$.

**Table 2.** Assignment of ecological resistance factors in this study.

| Ecological Resistance Factor | 1 | 2 | 3 | 4 | 5 |
|---|---|---|---|---|---|
| DEM | $\leq$75 | (75, 100] | (100, 125] | (125, 150] | $\geq$150 |
| Slope (°) | $\leq$5 | (5, 10] | (10, 20] | (20, 30] | $\geq$30 |
| LULC | Woodland | Grassland | Cropland | Water | Building land |
| Night lighting | $\leq$45 | (45, 50] | (50, 55] | (55, 60] | $\geq$60 |
| NDVI | >0.45 | (0.4, 0.45] | (0.35, 0.4] | (0.3, 0.35] | $\leq$0.3 |
| Population | $\leq$1000 | (1000, 3500] | (3500, 6000] | (6000, 8500] | $\geq$8500 |
| Distance to road (m) | >400 | (300, 400] | (200, 300] | (100, 200] | $\leq$100 |
| Distance to water source (m) | $\geq$600 | (450, 600] | (300, 450] | (150, 300] | $\leq$150 |
| Density of the road network | $\leq$450 | (450, 800] | (800, 1150] | (1150, 1500] | $\geq$1500 |
| Density of the water network | $\leq$5 | (5, 10] | (10, 15] | (15, 20] | $\geq$20 |

### 2.3.3. Principal Component Analysis

Principal component analysis (PCA) is a multivariate statistical method that is used to transform variables into independent principal components and is also a widely used network method. PCA transforms the values of a set of linearly uncorrelated variables, known as PCs, into the observations of a set of potentially correlated variables via an orthogonal transformation. In this research, weights were calculated using a weighted superposition approach based on PCA by thresholding how similar each data trace was to a reference trace created using PCA [26]. Since there are as many principal components in the data as there are variables, the principal components are constructed in such a way that the first component occupies the maximum possible variance in the set. The primary benefits of adopting the method of PCA include highlighting dataset commonalities, removing correlations between assessment metrics, drastically reducing the amount of metric selection and computation, and being a potent tool for dimensionality reduction when working with many datasets.

### 2.3.4. Multiple Scenario Simulation

The habitats selected here were mainly grassland, woodland, and unused land. With urban expansion and human activity, habitats have been decreased in area and become fragmented. As a comprehensive regional development activity, the development of unused land may cause certain ecological impacts on the development area and its surroundings, which requires advanced planning of the unused land development process. Thus, the unused land in the study area will change relatively drastically in the course of future urban development. Based on this, we simulated the unused land to provide advice for policy making and urban planning.

According to urban reshaping, large areas of unused land in the study area will transformed into residential areas, while in general, the green area ratio in new residential areas should not be less than 30%. In this study, the unused land was selected according to the urban development plan of Nanning City (Figure 1), and consists of areas that will be developed into human communities in the future. For the utilization of urban unused land, we put forth the A, B, C, and D scenarios for a human community (Figure 3). Scenario A divides the parcel into strips and selected several strips with 30% of the total area to be built as green land, while the rest are building land. Scenario B directly divide the parcel into two parts, where one part with 30% of the total area is green land and the other part is building land. In Scenario C, the central area of the plot is planned as a quadrangle, and 30% of the total area of this central area is green space. Scenario D is the opposite of Scenario C. The

peripheral area is 30% of the total area, which is green space, and the central area is 70% of the total area, which is building land. The unused land selected in the aforementioned areas was modified in the four different scenarios, but each scenario guarantees 30% green space, as shown in Figure 3. Lastly, the conversion tool in ArcGIS 10.8 was used to convert the 10 m land use data into vector files, which were then adjusted by type with cutting and splitting procedures to obtain the four processes of habitat fragmentation.

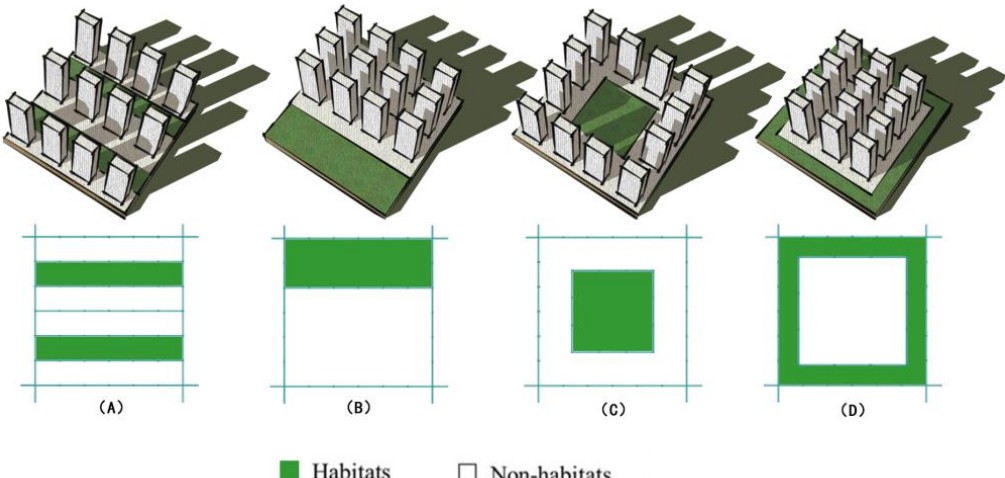

**Figure 3.** Schematic diagram of multi-scenarios simulations. These simulation scenarios in this study were selected regular unused land and modified into green land and unused land in the ratio of 3:7. Scenario A divided the parcel into strips. Scenario B divided the parcel into two strips. Scenario C planned the central area of the parcel as a quadrangle. Scenario D was the opposite of Scenario C, the peripheral area was green space. All scenarios of human communities that guarantee a 30% ratio of green space.

The conversion of unused land leads to changes in the original land cover, and thus data such as land cover and NDVI need to be adjusted. By combining the green land and building land obtained in the previous steps and replacing the original unused land, the new raster images of land cover were obtained using ArcGIS 10.8. Notably, the NDVI data were modified by extracting the modified unused land portion from the original data and reducing the NDVI by 70%. The habitats were then selected again, and the unused land that had been converted to building land was removed. The resistance surface was constructed by combining the resistance factors and calculating the weights. In accordance with the four different scenarios, the aforementioned operation was repeated.

### 2.3.5. Topological Indicators for Evaluating HCNs

The topological structure of a network illustrates how the nodes are connected to one another and how the network is arranged and configured. The spatial distribution of this environment resembles an undirected, unprivileged network when habitats are seen as nodes and corridors as edges. The topological relationships of HCNs reflect the overall spatial structure of habitats [16]. In this study, the HCNs were described using six topological indicators, including average degree, diameter, modularity, clustering coefficient, eigenvector centrality, and average path length [19,27,28].

#### Average Degree

The number of edges in a network that are connected to a node determines its degree, which reflects how connected a habitat patch is to its nearby patches in the HCN. The average degree of the network is determined as the average of all nodes' degrees [19,28]. The specific formula is as follows:

$$\langle k \rangle = \frac{2M}{N} \tag{2}$$

where $M$ is the number of network edges and $N$ is the number of nodes in the network.

Diameter

The diameter is the maximum distance between nodes $i$ and $j$ in the network. The connectivity between habitats can be shown by the diameter [19,28]. The specific formula is as follows:

$$D = max(d_{ij}) \tag{3}$$

where $D$ is the diameter and $d_{ij}$ is the distance between nodes $i$ and $j$.

Modularity

Modularity measures the degree of modularity of the network structure, and its value usually lies between 0.3 and 0.7. Usually, a modularity value greater than 0.44 indicates that the network has reached a certain degree of modularity. Modularity actually refers to the distinction between a network that is part of a certain community division and a random network, which means that a larger the difference means that the community division is better and the internal link density is higher [28,29]. The specific formulas are as follows:

$$Tre = \sum_i e_{ii} \tag{4}$$

$$a_i = \sum_j e_{ij} \tag{5}$$

$$Q = \sum_i (e_{ii} - a_i^2) \tag{6}$$

where $Q$ is the modularity and defines a $k \times k$ symmetric matrix $e = (e_{ii})$, where the element $e_{ij}$ denotes the proportion of edges in the network connecting nodes of two different associations among all edges; these two nodes are located in the $i$-th association and the $j$-th association, respectively.

Clustering Coefficient

The degree of connectedness between nearby nodes of a given node is represented by the clustering coefficient. The probability that two nodes connected to the same node will remain connected is known as the clustering coefficient of a node [19,28]. The specific formula is as follows:

$$C_a = \frac{E_a}{C_{k_a}^2} \tag{7}$$

where $C_a$ is the clustering coefficient of node $a$, $E_a$ is the actual number of edges between node $a$ and its neighboring nodes, and $C_{k_a}^2$ is the total number of edges when node $a$ and its neighboring nodes are connected to each other.

Eigenvector Centrality

The importance of a node is determined by the quantity and importance of its nearby nodes. In graph theory, eigenvalues are a way to measure the impact of a node on the network. For nodes with the same number of connections, a node with a high neighboring node score will score higher than a node with a low neighboring node score, and all nodes are assigned a corresponding score based on this principle [7,19,28]. A node connected to many other nodes that are connected to high-scoring nodes also has a high eigenvector score. The specific formula is as follows:

$$Ax = \lambda x \tag{8}$$

where each specific vector in the above equation corresponds to a different specific value $\lambda$, and each component of the eigenvector must be positive. According to the Perron–

Frobenius theorem, only the eigenvector corresponding to the largest eigenvalue is needed to measure centrality. To find this eigenvector, a power operation iterative algorithm can be used. The *i*-th component $x_i$ is the eigenvector centrality $C_E(v_i)$ of node $v_i$.

Average Path Length

The shortest distance between node *i* and node *j* is the path that connects these two nodes with the least number of edges. The average path length, which is the average of the distance between any two nodes, indicates how far apart the nodes in the network are from one another. A smaller value represents a greater connectivity of the nodes in the network [7,19]. The specific formula is as follows:

$$L = \frac{1}{\frac{1}{2}N(N-1)} \sum_{i>=j} d_{ij} \tag{9}$$

where $L$ is the average path length, $N$ is the number of nodes in the network, and $d_{ij}$ is the distance between node *i* and node *j*.

2.3.6. Robustness for Evaluating HCNs

Connectivity robustness and recovery robustness are two types of robustness that describe the capacity of networks to preserve their initial functionality in the face of uncertainties such external perturbations or internal parameter uptake [17,19,22–25]. The literature shows that patches and edges can be used to measure habitat fragmentation, reflecting the characteristics of changing landscape patterns in different habitats [30]. Connectivity robustness measures how well the remaining nodes can maintain connectivity even in the event that some nodes of the network are damaged. The ability of a network to recover when some of its components are disrupted is measured by its recovery robustness. The specific formulas are as follows:

$$R = \frac{C}{(N - N_r)} \tag{10}$$

$$D = 1 - \left[ \frac{(N_r - N_d)}{N} \right] \tag{11}$$

$$E = 1 - \left[ \frac{(M_r - M_e)}{M} \right] \tag{12}$$

where $N$ is the total number of nodes in the initial network, $C$ is the number of nodes in the maximum connected subgraph after eliminating $N_r$ nodes from the network, and $R$ denotes the connectivity robustness. The number of nodes recovered by a certain approach is $N_d$, and $D$ is the robustness of node recovery. Edge recovery robustness is measured by the letters $E$, $M_r$, $M_e$, and $M$, where $M$ is the total number of edges in the initial network, $M_e$ is the number of edges recovered by some approach, and $M_e$ is the number of edges deleted from the network.

## 3. Results

### 3.1. Spatial Distribution of Habitats

In this study, we investigated the impact of urban expansion on HCNs from both local and overall viewpoints. To conduct local network analyses, three major concentrations of the western, central, and eastern areas with strong land use changes were selected for urban reshaping (Figure 1).

Currently, habitats in the study area are relatively evenly distributed, with relatively large areas of concentrated woodland cover in two hills on both sides of the Yongjiang River in the southeast. The wetlands adjacent to the Yongjiang River have some adjacent grass and shrub patches. Overall, the land cover of habitats is dominated by woodland

and unused land. According to the real circumstances in the study area, a total habitat area of 153.24 km$^2$ was obtained, accounting for 34.08% of the study area. Figure 4 shows the spatial distribution of habitats.

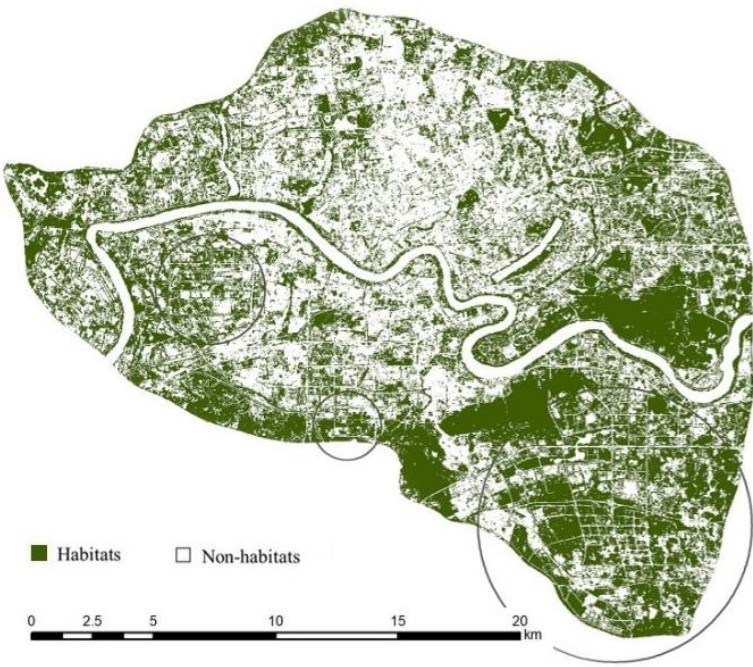

**Figure 4.** Spatial distribution of habitats, which includes woodland, grassland, and unused land.

After urban reshaping, habitats that are mainly influenced by land conversion of unused land reached 92.28 km$^2$. As shown in Figure 5, the largest region of the intended reshaping habitats was in the east, followed by the middle and western regions. In this study, we selected 80 habitat patches that met the aforementioned requirements. After simple treatment, 89 habitat patches were obtained. These 89 patches were modified in four different scenarios in turn to obtain 2106, 1873, 1886, and 1884 fragmented habitats. It can be seen that the spatial distribution of these habitats differed more significantly under different urban reshaping scenarios (Figure 5).

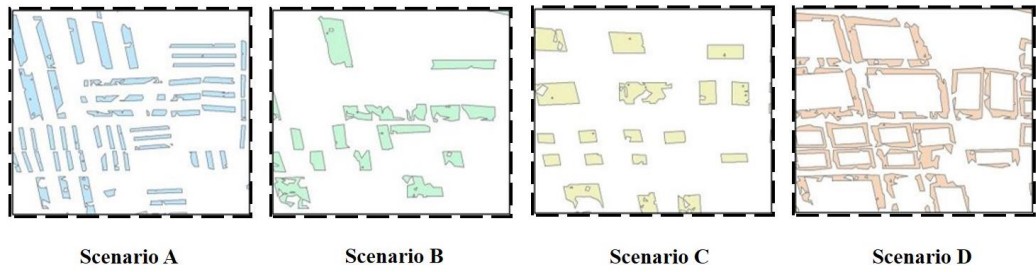

**Figure 5.** Spatial distribution of habitats under four different scenarios.

The local analysis contained 81 western patches, 35 central patches, and 278 eastern patches. It is worth noting that these three areas had higher concentrations of unused land. These three local areas were modified in different scenarios in turn, with 119, 87, 88 and 91 broken habitat patches in the western region, and 57, 41, 39 and 39 in the central part, and 502, 317, 331 and 326 in the eastern region, respectively. The fragmentation of habitats under Scenario A was significant during the planning and reshaping processes.

### 3.2. Principal Component Analysis and Minimum Cumulative Resistance Surface Analysis

Using ArcGIS 10.8, the resistance factors were reclassified and combined with principal component analysis to obtain the resistance surface. The weighted sum of the 10 resistance factors, which is the minimal cumulative resistance surface produced by the MCR model, ranged from 1.40 to 4.59. Due to more frequent human activities in the downtown area, worse environmental conditions, and a lower amount and quality of habitats, the cumulative effect of ecological resistance was more visible. The ecological function in the research region essentially rose from the center to the surrounding area, with the highest ecological function in the forest patches and unused land patches. All in all, the ecological resistance was strong in the central area and low in the surrounding area.

We conducted principal component analysis and constructed resistance surfaces for the study area under different scenarios. We found that the resistance values of the areas where unused land was converted were significantly higher, and the range of resistance values was nearly the same for all three categories except for Scenario C, which was lower. The conversion of unused land into residential areas increased the construction land by 70%, which also implied a significant reduction in habitats in the study area. Furthermore, we discovered that the regions with higher resistance values under the four scenarios were situated in roughly the same geospatial location, where the ecological environment was poor and the ecological resistance was high. And comparing the weights of resistance factors under the current situation and four simulation scenarios (Figure 6), it is easy to see the profound influence of weights. After urban reshaping, we found that the contribution of DEM, NDVI, road density, slope, and distance to water were elevated. In Scenarios A, B, C, and D, the difference in contribution rates was at most 0.1%, except for one indicator with the same contribution rate of distance from the road; the detailed information is shown in Figure 6.

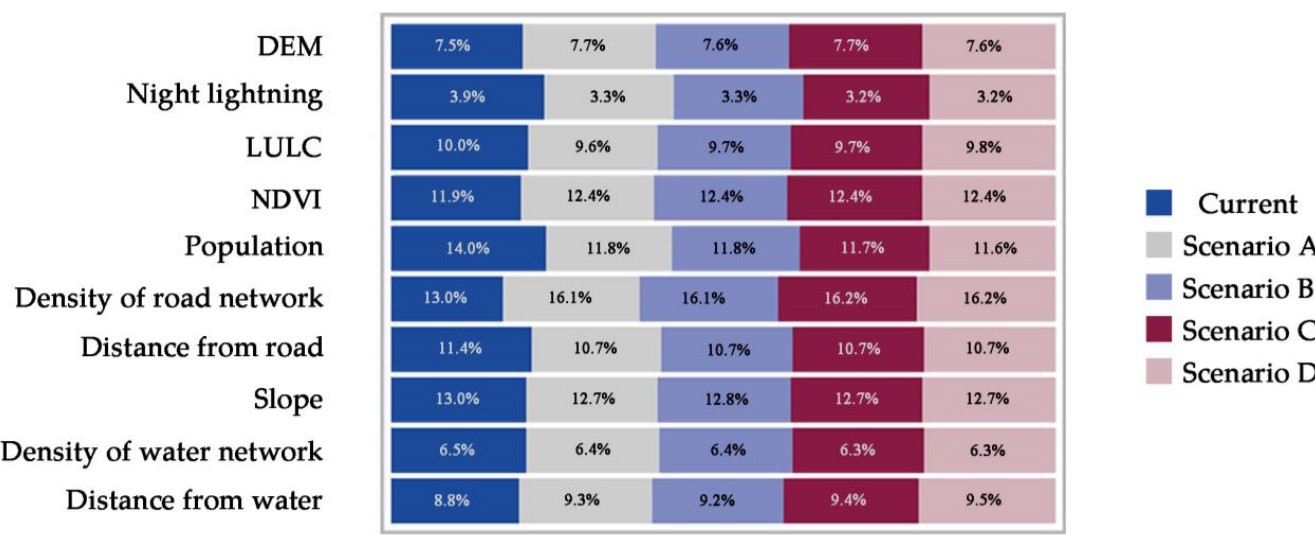

**Figure 6.** Comparison of principal component analysis weights.

### 3.3. Analysis of HCNs

Using the help of the Linkage Mapper tool in ArcGIS, we created HCNs for wildlife in the study area. Based on the network results, we obtained 5337 corridor lines in the current area, which are individually connected and densely distributed among them.

For the whole study area, the network results in different scenarios varied significantly; there were 6235, 5527, 5564, and 5548 ecological corridor lines, respectively (Table 3). Thus, the habitat connectivity priority was Scenario A > Scenario C > Scenario D > Scenario B. In addition, we discovered that the center of the study area had fewer and longer ecological corridors, and a sparser distribution of habitats than the outer regions, which had more ecological corridors and a more dense distribution of habitats. This shows that the ecological

position of the center is bad, the spatial structure of the landscape has to be modified, and urban expansion will have a negative impact on the habitats of wildlife.

**Table 3.** Habitat patches and ecological corridors of whole study area.

| Amount/Type | Current | Scenario A | Scenario B | Scenario C | Scenario D |
|---|---|---|---|---|---|
| Habitat patches | 1822 | 2106 | 1873 | 1886 | 1884 |
| Ecological corridors | 5337 | 6235 | 5527 | 5564 | 5548 |

As shown in Figure 7, a network analysis of the western, central, and eastern areas with significant local land use changes was conducted. A total of 766, 83, and 214 ecological corridors was obtained. Then, we conducted a multiple scenario simulation. The network results from the different scenarios yielded 1447, 893, 936, and 913 corridor lines in the east; 151, 101, 94, and 93 corridor lines in the central part; and 336, 238, 239, and 248 corridor lines in the west. The order of ecological effect priority is completely consistent with the simulation effect in the whole study area. Moreover, there was a general upward trend in the number of ecological corridor lines as habitat fragmentation increased; this change was similar to the upward trend in the number of habitat patches.

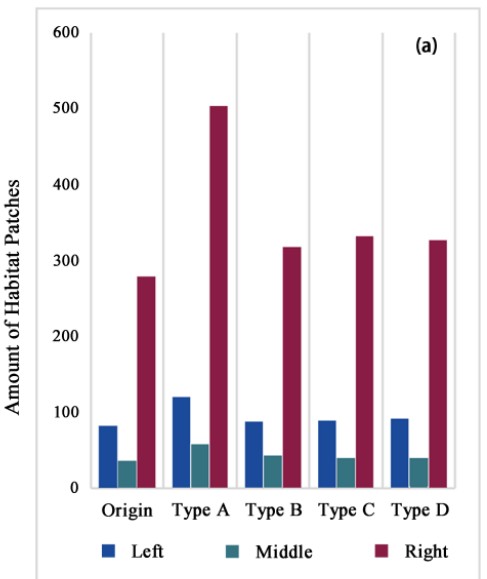 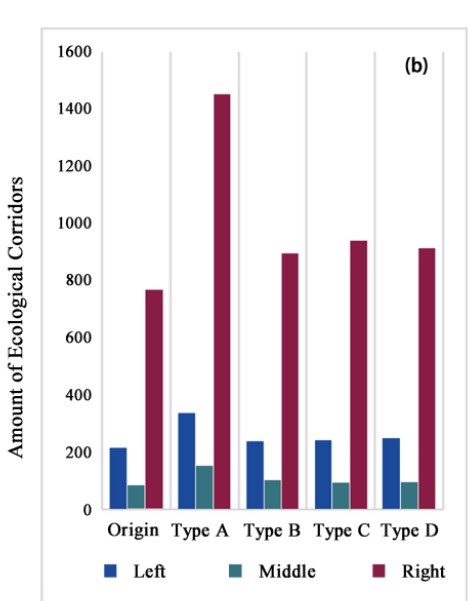

**Figure 7.** (**a**) Habitat patches of local study area, (**b**) ecological corridors of local study area.

### 3.4. Topological Indicators Analysis of HCNs

By visualizing the complex network of habitats, we obtained some results (Figure 8). In the entire study area, the greatest distance of in the HCN between habitat patches was 24, and the average path length between nodes was 10.598, requiring an average of 10.598 edges to be passed to connect. The connection relationships of nodes demonstrate that after reshaping, the average degree value and average path length grew, and the importance of nodes increased, with the largest increase in the average degree value in Scenario C and Scenario A. The eigenvector centrality became larger, among which, the eigenvector centrality of Scenario A increased greatly, indicating that the connectivity between nodes expanded and that the ability to transfer energy and information between nodes improved. In addition, the clustering coefficients decreased to some extent except in Scenario D, which remained basically stable. The definition of modularity states that nodes with high modularity have greater connectedness and resistance to natural and human-caused threats. The modularity of the future scenario increased slightly, indicating a somewhat higher linkage density within the network.

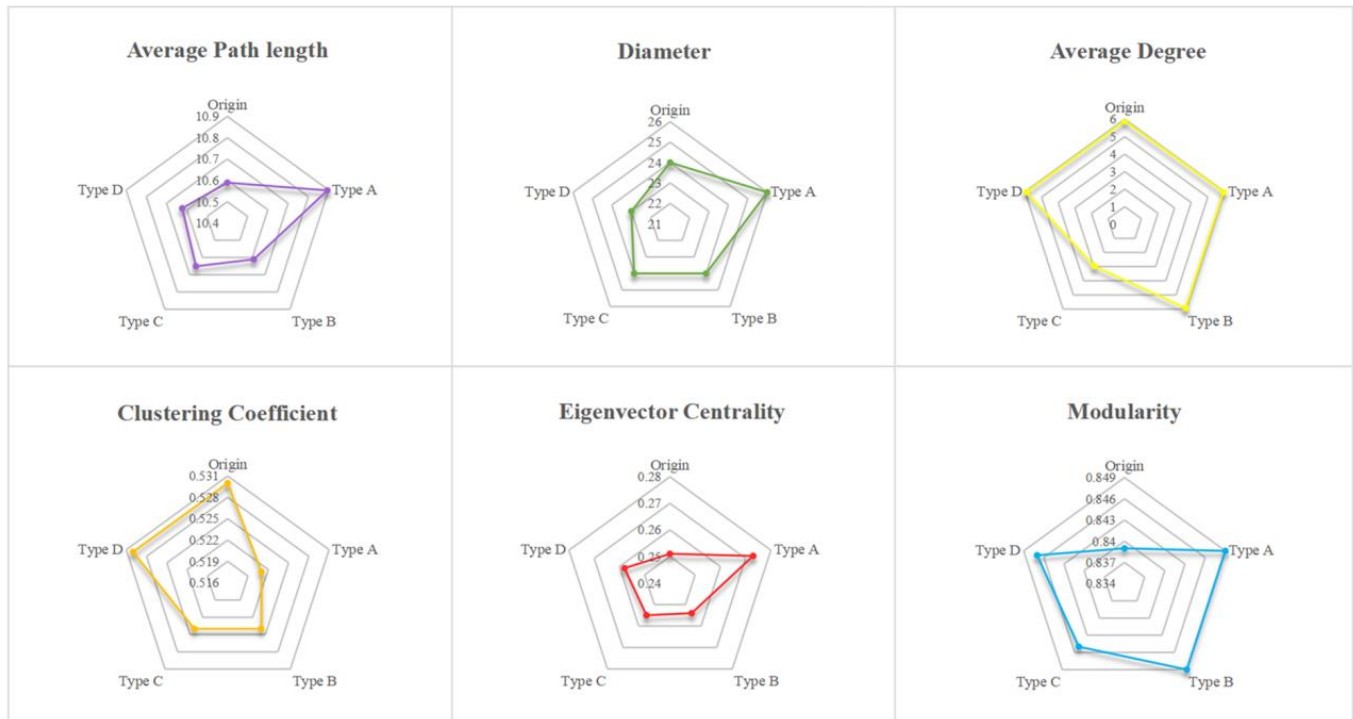

**Figure 8.** Topological indicators of HCNs for the entire study area.

For the three localized areas of the study area (western, central, and eastern), we calculated the topological values of each area's HCN with the HCNs under different scenarios (Table 4). We found that the results of the study were approximately the same as the analysis of the whole study area. The difference is that the graph density of the future scenarios were all lower than the graph density of the current study area, which means that the linkage density within the reshaping area was somewhat higher after urban reshaping.

**Table 4.** Topology indicators of habitat connectivity networks for the local study area.

| Topological Indicator | | Average Degree | Diameter | Modularity | Clustering Coefficient | Eigenvector Centrality | Average Path Length |
|---|---|---|---|---|---|---|---|
| Current | western | 5.284 | 9 | 0.585 | 0.523 | 0.012 | 3.884 |
| | central | 4.882 | 5 | 0.480 | 0.583 | 0.002 | 2.656 |
| | eastern | 5.551 | 12 | 0.718 | 0.544 | 0.036 | 5.405 |
| Scenario A | western | 5.647 | 9 | 0.648 | 0.508 | 0.018 | 4.301 |
| | central | 5.393 | 6 | 0.529 | 0.541 | 0.005 | 3.003 |
| | eastern | 5.776 | 14 | 0.763 | 0.515 | 0.069 | 6.347 |
| Scenario B | western | 2.736 | 9 | 0.617 | 0.509 | 0.011 | 4.040 |
| | central | 5.050 | 6 | 0.497 | 0.568 | 0.004 | 2.903 |
| | eastern | 5.652 | 12 | 0.723 | 0.523 | 0.044 | 5.69 |
| Scenario C | western | 5.432 | 9 | 0.615 | 0.525 | 0.011 | 4.016 |
| | central | 4.947 | 6 | 0.480 | 0.570 | 0.004 | 2.898 |
| | eastern | 5.673 | 13 | 0.722 | 0.518 | 0.046 | 5.839 |
| Scenario D | western | 5.451 | 9 | 0.621 | 0.533 | 0.013 | 4.021 |
| | central | 4.895 | 6 | 0.474 | 0.603 | 0.003 | 2.770 |
| | eastern | 5.618 | 12 | 0.736 | 0.541 | 0.043 | 5.627 |

*3.5. Robustness Analysis of HCNs*

We simulated robustness under the current state and scenario simulations, respectively, using random and malicious attacks. Connectivity robustness, edge recovery robustness, and node recovery robustness were the three types of robustness we focused on in this research. The results show that the robustness against malicious attacks decreased faster than that of random attacks regardless of the simulation scenario. In addition, the decreasing trend of recovery robustness approximated a convex curve, decreasing slower at the beginning of the attack than at the end of the attack, while the decreasing trend of edge recovery robustness under malicious attacks decreased linearly. In addition, the decreasing trend of connectivity robustness approximated a concave curve, increasing with the rate of decline and then slowing down with the attack intensity.

The three robustness results of the networks in the present and future Scenarios A, B, C, and D (Figure 9) show that urban expansion weakens the stability of the network of habitats, with the same conclusions obtained in the overall and local analysis. Overall, Scenario D showed the best results for urban expansion. We discovered that its robustness in terms of connectivity was distinct from the other scenarios, and the fall in connectivity robustness displayed a wave-like decline. After simulating future scenarios in the study area, the robustness of the HCNs slightly decreased after malicious and random attacks. The decreasing connectivity robustness curve for the status quo was more concave during malicious attacks than it was after simulation, but the inflection point was delayed. In the HCNs under all scenarios simulated, the decline in connectivity robustness and the onset of network connectivity collapse were advanced. As shown in Figure 9, when 450 nodes were maliciously attacked, the robustness of the status quo fell below 0.1 and the network crashed, while the network robustness of the future scenario with less than 450 nodes maliciously attacked fell below 0.1. In the random attack, the robustness of the network in the future scenario decreased rapidly as the number of attacked nodes increased, while the network of the status quo still maintained a high robustness and decreased slowly. Regarding the recovery robustness of the nodes, the beginning decline point, curve inflection point, and network collapse point for the future scenario were all moved to the right under both attack modes compared to the pre-simulation period, indicating a later appearance. The network robustness of the status quo stayed above 0.9 after 1160 and 1225 nodes were attacked and dropped below 0.1 and eventually crashed after 1730 and 1800 nodes were attacked. In addition, the node recovery robustness curve of the status quo was somewhat more convex in the late attack period than in the future scenario. Both the current situation and the future scenarios showed no significant change in the initial decline in recovery robustness for both attack modes, but the curve inflection point and the network crash point moved to the right, and the status quo was slightly more convex than after the simulation. The robustness of the network in future scenarios started to decline during the initial attack, while the current network robustness continued to be very high. However, in the subsequent attacks, the robustness declined at approximately the same rate. Under both attacks, the status quo about 1800 edges needed to be attacked, while in the future scenario needed to have more than 1800 edges attacked to bring the robustness of recovery down to 0. Analyzing the study area locally and combining the conclusions obtained from the above analysis of the study area as a whole, we concluded that in terms of HCN stability and ecological resilience, the ranking is Scenario D > Scenario B ≈ Scenario C > Scenario A.

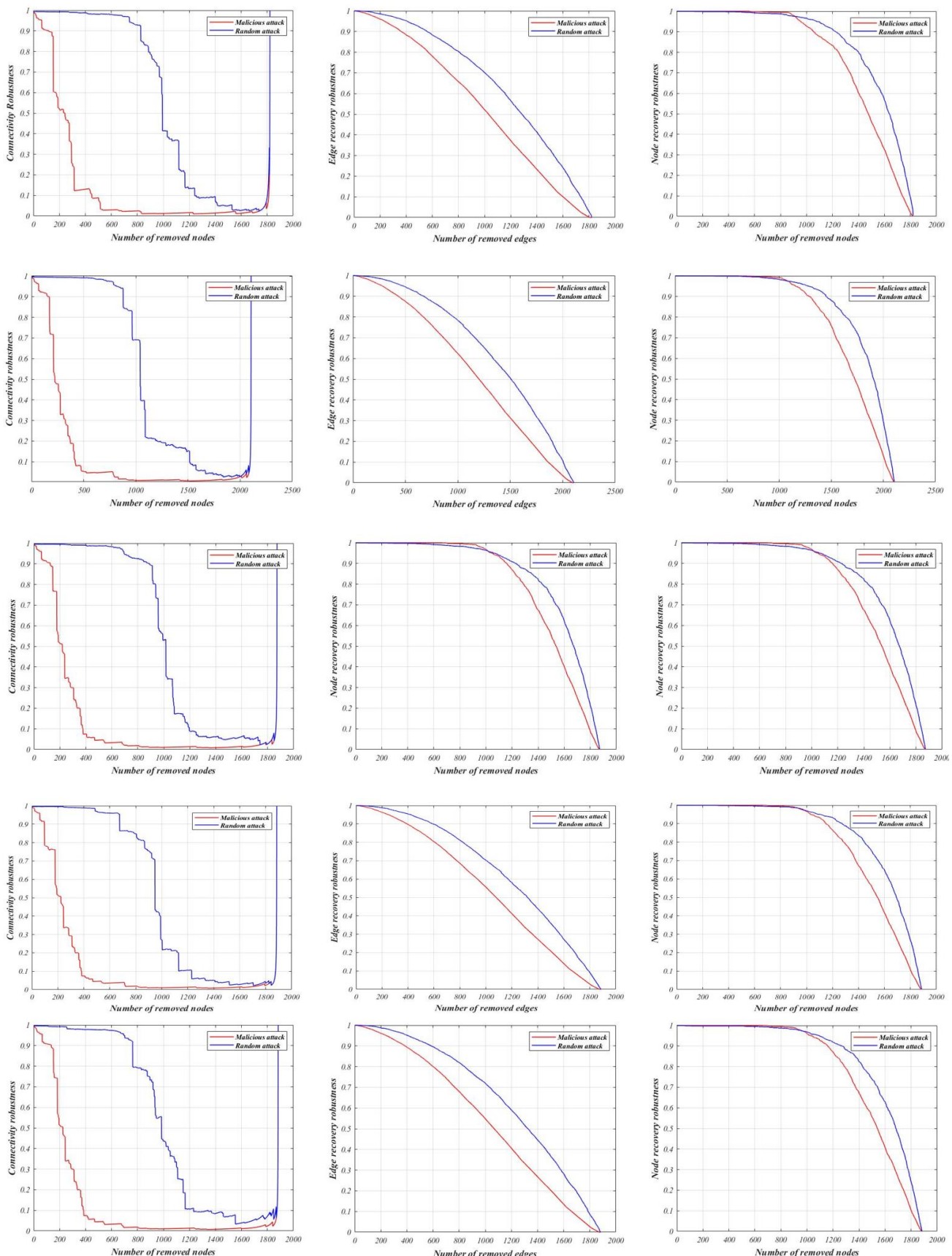

**Figure 9.** Robustness of HCNs for the whole study area. One scenario per row, from **top** to **bottom**: current, Scenario A, Scenario B, Scenario C, Scenario D.

## 4. Discussion

### 4.1. Construction of HCNs

Currently, wildlife conservation and habitat networking are important components of global spatial planning, and improving the connectivity of habitat networks has important implications. The research has started to concentrate on networks of habitat connection during the planning and reshaping process as a result of urbanization [31–34]. Connectivity between habitats is essential to maintaining communication between biological populations, and communication between species can be facilitated by improving the connectivity of habitat networks [33]. This emphasizes the importance of constructing HCNs. As a result, many researchers have explored different approaches to build biological habitat networks. Hofman proposed a habitat network assessment framework using habitat suitability indicators and graph theory, and applied it to biological habitat network construction [35]. Changes in urban land use can affect habitat connectivity, particularly in the context of urban expansion. The structure analysis of urban HCNs based on the minimum cumulative resistance model has received more attention recently [36,37]. However, most studies on HCNs have mainly focused on the assessment of the current situation, while very few have applied the results of network analysis to construct HCNs and predict future patterns and determine priority interventions.

Compared with previous studies, our research is unique and future facing. Our study developed and evaluated HCNs during urban expansion, comparing the results of the HCNs under current and future scenarios. Importantly, our study presents four possible scenarios (Figure 3) that are able to demonstrate the inevitability of future urban expansion and assess its impact on future development to support decisions on urban planning and wildlife conservation. Why were only these four scenarios considered? Because human communities typically expand in a more intensive manner along with population growth. We gathered four of the more typical scenarios (Scenarios A, B, C, and D) by consulting numerous development cases. Of course, there are many other scenarios out there waiting to be explored.

### 4.2. Urban Expansion and the HCNs

Urban expansion has accelerated recently in developing countries like China. The research has shown that habitat quality is negatively correlated with the level of urbanization [1,5,6,38,39]. The study area is experiencing rapid urban expansion with a gradually expanding urbanized core and a gradual replacement of adjacent natural land cover by built-up land [40]. According one study, urban expansion has led to habitat loss, habitat fragmentation, and a decline in diversity of wildlife [41]. Urban expansion may have a negative impact on wildlife diversity because of the alterations it causes to natural habitats rather than because of the urban expansion itself. As opposed to earlier research that examined the relationship between habitat and urbanization, our results suggest that changes in the distribution of different land cover types during urban development are directly related to the HCNs for wildlife in urban areas. Exploring the complex effects of habitat fragmentation on wildlife habitats due to urban expansion and identifying specific outcomes triggered by fragmentation processes are important issues for relevant scientific research and urban planning [42–44].

Urban expansion has weakened the stability of HCNs, posing a serious ecological challenge to the future planning and construction of cities. In contrast, the negative impacts of urban expansion can be minimized through appropriate urban development planning and ecological restoration. This study examined the construction of HCNs and their evolution through urban expansion, with results that differ from studies that have mainly used a simple analysis or non-quantitative methods. Using tools such as ArcGIS and Matlab to analyze the constructed HCNs, the topological indicators and robustness curves of the four scenarios of urban reshaping indicate that urban expansion weakens the stability of HCNs. Moreover, the four scenarios and their simulation results were selected and differentiated. For example, the individual patches in Scenario A are small and dispersed, despite the fact

that the total habitat area was the same in each scenario. Ultimately, we concluded that the four scenarios were ranked as follows: Scenario D > Scenario B ≈ Scenario C > Scenario A. These results quantitatively describe the complex effects of different scenarios of urban reshaping on wildlife habitats and reveal the importance of further research on the relationship between urban planning and wildlife conservation. Based on the results of this paper, we suggest that Scenario D be chosen for urban residential land use planning for the reshaping of unused land to better maintain the connectivity of habitats.

### 4.3. Implications for Management

Cities perform as an evolutionary system, continuously disrupting natural landscapes [45,46]. Research on land-use change in the past has shown how urban land expansion affects natural environments [47–51]. Nonetheless, urbanization is accelerating and developing at an unprecedented rate globally, which is displacing unused land, cropland, and woodland in the areas surrounding big cities [47,48]. Our study shows that urban expansion has some effects on formerly stable environments that are in danger of environmental degradation. This may not be restricted to the destabilization of environmental structures and destruction, and may include a decline in habitat quantity and quality. This entails reducing urban expansion while ensuring development. Additionally, we carried out this study in local and overall perspectives, and built HCNs for the entire study area as well as for each of the three locations with significant land use changes. The findings demonstrate that urban expansion has a larger detrimental effect on local HCNs and may potentially be destructive. Thus, we must give ecological conservation in localized areas more consideration.

Furthermore, we developed four scenarios for human communities that mostly rely on unused land (planned for human communities). Among the four scenarios, we discovered that Scenario D maximizes the demand for urban expansion, which ensures that the needs of a growing population for community development are met while at the same time, mitigating environmental degradation. So when urbanization is inevitable, using Scenario D to construct human communities in similar urban areas can better safeguard natural habitats. Our research has important implications for both the planning of human communities and the preservation of urban ecological patterns.

### 4.4. Limitations and Future Research Directions

Although our results provide insights for future urban planning and wildlife conservation, this study has its limitations. Firstly, there are no uniform criteria for the selection of ecological resistance coefficients, and differences in the resistance coefficients in the same study area may result in different HCNs. Secondly, we only studied the migration of wildlife in the horizontal dimension of the region, without considering the important influences in the vertical direction or variations in species movement based on seasons. This leads to some restrictions in the assessment of the HCNs. Thirdly, the choice of urban expansion and reshaping scenarios is worth discussing. In our study, we chose four reshaping scenarios for residential land. However, there are many other scenarios, or better results than those obtained from the four development scenarios mentioned in this paper. Finally, the construction and simulation of HCNs is mainly to provide guidance for future urban planning and wildlife conservation, but it did not take into account the socio-economic factors. For example, how to maximize ecological benefits within limited economic resources is a question that deserves further exploration.

In the future, we will overcome the existing limitations, and set up thorough evaluation and analysis methods that includes both vertical and horizontal directions. We will screen more possible scenarios to reshape unused land and conduct more case studies in different areas or at different scales to enrich our conclusions. Furthermore, a broad range of wildlife species exist, each with distinct habitat requirements and migratory routes. To obtain more general results, multi-species studies will be performed. Alvey highlights that urban

planners should be aware that cities provide a foothold for biodiversity in urban areas [52]. Thus, it is also necessary to incorporate ecologists in the planning of urban expansion.

## 5. Conclusions

Urban expansion is an inevitable trend in human development. People are hopeful that, through scientific urban planning and the adoption of innovative models for human communities, such a situation can be improved. Hence we proposed the A, B, C, and D scenarios of human communities, given the global urbanization trend and the increasing conflicts with wildlife. Comparing the results of the four scenarios allows us to identify more appropriate development scenarios to minimize the negative consequences of urban expansion. In this study, we chose the habitat of Nanning City in Guangxi Province of southwest China. We built an ecological resistance surface based on circuit theory, principal component analysis, complex network, etc. Then, the fragmented habitats and ecological corridors were used to composed the HCNs. It was discovered that the process of urban expansion has a detrimental effect on wildlife habitats. This is shown by the 64.60 km$^2$ decrease in habitats, the disruption of the HCNs' structure, and the decline in environmental stability. Also, the effects of various urban reshaping scenarios on the alterations in wildlife habitats were very distinct, and their respective network stability was compromised to varying degrees. In contrast, the topological indicators and robustness effects of Scenario D had less negative impacts under expansion. It is important to note that Scenario D refers to the situation where the quadrilateral of the plot's central region is changed into construction land and is surrounded by grassland during the creation of the human communities. It was inferred that urban planning using Scenario D is the most desirable development option for better protection of wildlife habitats.

Furthermore, Scenario D can be extended to similar human communities development processes and is not limited to the specific case study under consideration. Although we have chosen the habitats of Nanning to study the impact of urban expansion on HCNs of urban wildlife, there are also implications for other urban wildlife network studies. Accordingly, we can mitigate the damage to wildlife caused by urban expansion. In conclusion, we provided a fresh perspective for similar case studies in urban areas, providing theoretical support and insights for further applications in urban planning and wildlife conservation in urban areas. In further studies, more urban reshaping scenarios and more reasonable evaluation methods can to be selected to provide more possibilities and improve the accuracy of the results, providing more information for understanding the impacts of urban expansion on wildlife habitats. This will assist with urban planning decisions and policy-making.

**Author Contributions:** S.C.: Conceptualization, Methodology, Software, Field experiments, Formal Analysis, and Writing—Original Draft; K.S.: Funding acquisition and Writing—review and editing; X.J.: Writing—review and editing; Y.Y.: Writing—review and editing; C.L.: Data pre-processing collation; L.W.: Data pre-processing collation. All authors have read and agreed to the published version of the manuscript.

**Funding:** This work was supported by the Open Foundation of the State Key Laboratory of Urban and Regional Ecology of China (SKLURE2023-2-3) and the Youth Science Foundation of Guangxi Province (2022GCXNSFBA035570).

**Data Availability Statement:** The authors do not have permission to share the data.

**Acknowledgments:** For their support in this study, we are thankful to the Guangxi Colleges and Universities Key Laboratory for Cultivation and Utilization of Subtropical Forest Plantation, College of Forestry, Guangxi University, and the State Key Laboratory of Urban and Regional Ecology, Research Center for Eco-Environmental Sciences, Chinese Academy of Sciences.

**Conflicts of Interest:** The authors declare that they have no known competing financial interests or personal relationships that could have appeared to influence the work reported in this paper.

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
