# Peer review of "Impacts and Predictions of Urban Expansion on Habitat Connectivity Networks: A Multi-Scenario Simulation Approach"

_forests, doi:10.3390/f14112187_

Round 1

Reviewer 1 Report

Comments and Suggestions for Authors

Manuscript title:  Impacts and predictions of urban expansion on habitat connectivity network: A multi-scenarios simulation approach

This study aims to comprehensively assess the future impact of land use, differentiating between used and unused land. It examines four scenarios of human communities and establishes a relationship with habitat connectivity networks in the southern China region. I truly appreciate the significance of conducting this study, especially in light of the current global population growth. Such research provides valuable insights into the consequences of urban expansion.

I commend the authors for their well-structured and formatted paper. Reviewing this paper has been a pleasant experience, and I believe it is well-prepared for publication with only minor corrections required. The flow of the paper is well-organized and effectively conveys its message. Authors diligent work on this study is highly appreciated.

Please provide quantitative values when referencing the results in the abstract and conclusion.

Avoid repetitive use of the full form of acronyms.

Consider reducing the size of the captions for Figure 3.

Regarding the choice of scenarios, it would be beneficial to explain why only these four scenarios were considered and if there were specific reasons for not including others.

Please elaborate on the authors' confidence in their results and provide the rationale behind their confidence level.

Author Response

"Impacts and predictions of urban expansion on habitat connectivity network: A multi-scenarios simulation approach"

Response to Comments of Reviewer #1

We feel great thanks for your professional review work on our article. As you are concerned, there are several problems that need to be addressed. According to your nice suggestions, we have made some corrections to our previous draft, the detailed corrections are listed below:

Comment 1:

Please provide quantitative values when referencing the results in the abstract and conclusion.

Response 1:

Thank you for your suggestions.

We added “This is shown by the 64.60 km2 decrease in habitats, the disruption of the HCNs' structure, the decline in environmental stability” to conlusion.

The text has too many relevant topological metrics and robustness, and the characters of the abstract limit us to a more detailed description.

Thank you and your best understanding.

Comment 2:

Avoid repetitive use of the full form of acronyms.

Response 2:

Thank you for your careful checks. We made the acronyms in the text as standardised as possible.

Comment 3:

Consider reducing the size of the captions for Figure 3.

Response 3:

We think this is an excellent suggestion. 

We have moved “ Scenario A divided the parcel into strips and selectd several strips with 30% of the total area to be built as green land, while the rest were building land. Scenario B directly divided the parcel into two parts, where one part with 30% of the total area was green land and the other part was building land. Scenario C divided the central area of the plot was planned as a quadrangle, and 30% of the total area of this central area was green space. Scenario D was the opposite of Scenario C. The peripheral area was 30% of the total area, which was green space, and the central area was 70% of the total area, which was building land” from Figure name to text. The length of the figure name is now shorter and the image size has been reduced.

Comment 4:

Regarding the choice of scenarios, it would be beneficial to explain why only these four scenarios were considered and if there were specific reasons for not including others.

Response 4:

Thank you very much for yourcomments andprofessional advice.

Regarding your question, we have explained it in the discussion 4.1 of the article. Such for “As for why only these four scenarios were considered? Because human communities typically expand in a more intensive manner along with population growth. We gathered four of the more typical scenarios (Scenario A, B, C, and D) by consulting numerous development cases. Of course, there are a ton of other scenarios out there waiting to be explored that might not be representative enough”. (line 539)

We understand your concerns very well, and feel great thanks for your professional review. 

Comment 5:

Please elaborate on the authors' confidence in their results and provide the rationale behind their confidence level.

Response 5:

We sincerely appreciate the valuable comments.

At your suggestion we have added a subsection (4.3) to the discussion section which we hope will satisfy you. The specific content are as follows:

“Cities perform as an evolutionary system, disrupting natural landscapes continuously [46,47]. Research on land-use change in the past has shown how urban land expansion affects natural environments [47,49-52]. Nonetheless, urbanization is accelerating and developing at an unprecedented rate globally, which is displacing unused land, cropland, and woodland in the areas surrounding big cities [48,49]. Our study shows that urban expansion has some effects on formerly stable environment that are at danger of environmental degradation. This may not be restricted to the destabilisation of environmental structures and destruction, as well as the decline in habitat quantity and quality. This entails reducing urban expansion while ensuring development. Additionally, we carried out this study in local and overall perspectives, and built HCNs for the entire study area as well as for each of the three locations with significant land use changes. The findings demonstrate that urban expansion has a larger detrimental effect on local HCNs and may potentially be destructive. Thus, we must give ecological conservation in localized areas more consideration.

Furthermore, we developed four scenarios for human communities that mostly rely on unused land (planned for human communities). Among the four common scenarios, we discovered that Scenario D maximizes the demand for urban expansion, which ensures that the needs of a growing population for community development are met while at the same time mitigating environmental degradation. So when urbanization is inevitable, using Scenario D to construct human communities in similar urban areas can better safeguard natural habitats. Our research has important implications for both the plan of human communities development and the preservation of urban ecological pattern.” (line 580-602)

Thanks again for your advice.

We tried our best to improve the manuscript and made some changes marked in red in the revised paper that will not influence the content or framework of the paper. We appreciate the reviewers’ earnest work and hope the correction will meet with approval. Once again, thank you very much for your comments and suggestions.

Reviewer 2 Report

Comments and Suggestions for Authors

Review:

The article titled "Impacts and predictions of urban expansion on habitat connectivity network: A multi-scenario simulation approach" provides valuable insights into the consequences of urban expansion on habitat connectivity networks and offers important guidance for urban planners aiming to mitigate the adverse effects of urban growth on wildlife conservation and ecosystem stability.

Praise the authors for their effort and contribution to the article. It is evident that they have put a lot of work into preparing this text. However, I have some comments and recommendations that could enhance the article.

ABSTRACT:

For improved readability, I recommend providing a brief description of the A, B, C, and D scenarios. Subsequent statements in the article refer to these scenarios, which cannot be understood without a comprehensive reeding of the whole article's content.

INTRODUCTION:

Please expand the background section and include more relevant references, including basic ones (e.g., Forman and Godron, McHarg, ...).

MATERIALS AND METHODS:

Regarding the term "unused land" (first mentioned in line 118), I recommend immediately providing an explanation of the type of non-use being referred to. Is it the same area as "Bareland" (Figure 2)? If so, why is there a different designation?

For "DEM" and "NDVI" (line 127), when using abbreviations for the first time, it's necessary to state their full names as indicated in lines 131 and 134.

Please also specify the reference of formulas 1-12.

RESULTS:

This chapter is somewhat challenging to read and lacks fluency. Readers are visual beings, and they are typically drawn to graphics, tables, and figures before diving into the interpretation. I recommend restructuring the entire chapter so that images are presented first, followed by the description of the results. This would greatly enhance the article's readability.

First, display the entire area and the three individual areas (Figure 6, line 358), and then show the spatial distribution of habitats (Figure 4, line 338).

Regarding the choice of area designation (Figure 4, Figure 6), there appears to be a dual nomenclature (e.g., left area, middle area, right area, or western, central, eastern). It would be better to opt for a consistent naming convention and use it consistently across all figures and graphs.

Please provide clarity on why the entire processing area was selected in some cases, as indicated in Figure 8, while in other instances, only the area within the circles was analyzed (e.g., Figure 9). In the interpretation of the results, it is essential to specify which area is included in each analysis and the rationale for this choice.

Could you explain why the areas labeled as left, middle, and right are of different sizes (as shown in Figure 6)? Are these truly circular areas, or do they follow the course of roads or parcel boundaries?

Additionally, it would be beneficial to clarify the significance of the red lines in the representation of the areas (Figure 6).

CONCLUSION:

It is crucial to clearly define the contribution to science. Additionally, it is important to explicitly state whether the results (e.g., the selection of Scenario D) are applicable solely to the specific case study or if they can be generalized to broader contexts.

Review conclusion: I recommend this article to be published after minor revision.

Author Response

"Impacts and predictions of urban expansion on habitat connectivity network: A multi-scenarios simulation approach"

Response to Comments of Reviewer #2

Thank you very much for your comments and professional advice. These opinions help to improve the academic rigor of our article. Based on your suggestion and request, we have made corrections to the revised manuscript. We hope that our work can be improved again. In the following, we will address each of these comments in detail.

Comment 1:

ABSTRACT:

For improved readability, I recommend providing a brief description of the A, B, C, and D scenarios. Subsequent statements in the article refer to these scenarios, which cannot be understood without a comprehensive reeding of the whole article's content.

Response 1:

We sincerely appreciate the valuable comments. We added a brief description of the A, B, C, and D scenarios. (line 21)

The current version is as follows, “To simulate changes to unused land in the future, we put forth the A (divided the parcel into some strips), B (divided the parcel into two strips), C (the central area of the parcel was planned as a quadrangle), and D (opposite of Scenario C, the peripheral area was green space) scenarios of human communities that guarantee a 30% ratio of green space, and established corresponding HCNs.” Due to space constraints, we can only give a short summary. Thank you for your understanding.

Comment 2:

INTRODUCTION:

Please expand the background section and include more relevant references, including basic ones (e.g., Forman and Godron, McHarg, ...).

Response 2:

We think this is an excellent suggestion. So we expanded the background including basic ones ( Forman and Godron). And added literature references elsewhere in the article mading our article more scientific.

The additions are as follows, “Ecological network aims to tackle the threats of habitat fragmentation on species' survival and migration, which have been the focus of ecological research since the 1980s [12,13,14]. Since Forman and Godron [14] proposed that landscape structure consists of three basic elements: patches, corridors, and matrix, the concept of ecological networks has been extended and developed, then its related exploration and research contribute to the maintenance of biodiversity and promote species dispersal, which has been widely used in academia.” (line 52)

Comment 3:

MATERIALS AND METHODS:

  • Regarding the term "unused land" (first mentioned in line 118), I recommend immediately providing an explanation of the type of non-use being referred to. Is it the same area as "Bareland" (Figure 2)? If so, why is there a different designation?

  • For "DEM" and "NDVI" (line 127), when using abbreviations for the first time, it's necessary to state their full names as indicated in lines 131 and 134.
  • Please also specify the reference of formulas 1-12.

Response 3:

  • We are sorry for our carelessness. We have aligned the relevant statements in the article, and have revised "Bareland"to "Unused land" in Figure 2.
  • We have modified the acronyms according to your request.
  • We think this is an excellent suggestion.So we specified the reference of formulas 1-12 as possible.

Comment 4:

RESULTS:

This chapter is somewhat challenging to read and lacks fluency. Readers are visual beings, and they are typically drawn to graphics, tables, and figures before diving into the interpretation. I recommend restructuring the entire chapter so that images are presented first, followed by the description of the results. This would greatly enhance the article's readability.

  • First, display the entire area and the three individual areas (Figure 6, line 358), and then show the spatial distribution of habitats (Figure 4, line 338).
  • Regarding the choice of area designation (Figure 4, Figure 6), there appears to be a dual nomenclature (e.g., left area, middle area, right area, or western, central, eastern). It would be better to opt for a consistent naming convention and use it consistently across all figures and graphs.
  • Please provide clarity on why the entire processing area was selected in some cases, as indicated in Figure 8, while in other instances, only the area within the circles was analyzed (e.g., Figure 9). In the interpretation of the results, it is essential to specify which area is included in each analysis and the rationale for this choice.
  • Could you explain why the areas labeled as left, middle, and right are of different sizes (as shown in Figure 6)? Are these truly circular areas, or do they follow the course of roads or parcel boundaries?
  • Additionally, it would be beneficial to clarify the significance of the red lines in the representation of the areas (Figure 6).

Response 4:

Thank you very much for yourcomments andprofessional advice.

  • We have represented the three localised regions in Figure1 (includes entire study area). 
  • We have opted for a consistent naming convention (western, central, eastern) and used it consistently across all figures and graphs. In particular, you can see the naming in Figure 1.
  • It is worth noting that we have analysed and described allareas (entire area and circles were fully processed and analysed), but there is an excessive amount of relevant data and graphs. So we have chosen a selection to show, thank you for your understanding. If you need our supplementary material, we will deliver it to you.
  • We concentrated on and made three circlesaround areas of significant land use change, which differ in area from one another because they were not initially consistently clustered. In order to differentiate between these circles, we named them western, central, eastern area. Moreover, they follow the course of roads or parcel boundaries.
  • I'm sorry for the trouble, these red lines only serve to highlight. To avoid such problems, I have labelled them directly in the research area(1, 2, 3 in Figure 1).

Comment 5:

CONCLUSION:

It is crucial to clearly define the contribution to science. Additionally, it is important to explicitly state whether the results (e.g., the selection of Scenario D) are applicable solely to the specific case study or if they can be generalized to broader contexts.

Response 5:

Thanks for your suggestion.

Regarding the research and contributions, we added some sentences in Conclusions. “Furthermore, Scenario D can be extended to similar human communities development processes and is not limited to the specific case study under consideration.” (line 650)

We tried our best to improve the manuscript and made some changes marked in red in the revised paper that will not influence the content or framework of the paper. We appreciate the reviewers’ earnest work and hope the correction will meet with approval. Once again, thank you very much for your comments and suggestions.
